# Targeting pathogenic CD8+ tissue-resident T cells with chimeric antigen receptor therapy in murine autoimmune cholangitis

Hao-Xian Zhu [1,2,10], Shu-Han Yang [1,10], Cai-Yue Gao[3], Zhen-Hua Bian[1], Xiao-Min Chen[1,2], Rong-Rong Huang[4], Qian-Li Meng[5], Xin Li[3], Haosheng Jin[6], Koichi Tsuneyama[7], Ying Han [8], Liang Li [3] ✉, Zhi-Bin Zhao [3] ✉, M. Eric Gershwin [9] ✉ & Zhe-Xiong Lian[2] ✉

Primary biliary cholangitis (PBC) is a cholestatic autoimmune liver disease characterized by autoreactive T cell response against intrahepatic small bile ducts. Here, we use Il12b⁻/⁻Il2ra⁻/⁻ mice (DKO mice) as a model of autoimmune cholangitis and demonstrate that *Cd8a* knockout or treatment with an anti-CD8α antibody prevents/reduces biliary immunopathology. Using single-cell RNA sequencing analysis, we identified CD8+ tissue-resident memory T (Trm) cells in the livers of DKO mice, which highly express activation- and cytotoxicity-associated markers and induce apoptosis of bile duct epithelial cells. Liver CD8+ Trm cells also upregulate the expression of several immune checkpoint molecules, including PD-1. We describe the development of a chimeric antigen receptor to target PD-1-expressing CD8+ Trm cells. Treatment of DKO mice with PD-1-targeting CAR-T cells selectively depleted liver CD8+ Trm cells and alleviated autoimmune cholangitis. Our work highlights the pathogenic role of CD8+ Trm cells and the potential therapeutic usage of PD-1-targeting CAR-T cells.

Primary biliary cholangitis (PBC) is a chronic autoimmune liver disease characterized by progressive intrahepatic small bile duct injury and elevated anti-mitochondrial antibodies (AMA)[1]. The pathogenesis of this disease involves loss of tolerance to the E2 subunit of the mitochondrial pyruvate dehydrogenase complex (PDC-E2) and autoimmune injury of biliary epithelial cells (BECs), resulting in cholestasis and progressive liver fibrosis[2]. The increased infiltration of CD4+ T and

CD8+ T cells in the liver indicates that abnormally activated T cells are involved in BEC apoptosis[3,4]. However, the mechanisms of immune dysregulation remain to be identified for potential therapeutic strategies. Approximately 20–40% of PBC patients do not fully respond to ursodeoxycholic acid (UDCA), which is currently the first-line drug approved for the treatment of PBC[5]. We previously developed and optimized several PBC murine models to investigate pathogenic

[1]Chronic Disease Laboratory, Guangdong Provincial People's Hospital (Guangdong Academy of Medical Sciences), School of Medicine, South China University of Technology, Guangzhou, China. [2]Guangdong Provincial People's Hospital (Guangdong Academy of Medical Sciences), Southern Medical University, Guangzhou, China. [3]Medical Research Institute, Guangdong Provincial People's Hospital (Guangdong Academy of Medical Sciences), Southern Medical University, Guangzhou, China. [4]Guangdong Cardiovascular Institute, Guangdong Provincial People's Hospital (Guangdong Academy of Medical Sciences), Southern Medical University, Guangzhou, China. [5]Guangdong Eye Institute, Department of Ophthalmology, Guangdong Provincial People's Hospital (Guangdong Academy of Medical Sciences), Southern Medical University, Guangzhou, China. [6]Department of General Surgery, Guangdong Provincial People's Hospital (Guangdong Academy of Medical Sciences), Southern Medical University, Guangzhou, China. [7]Department of Pathology and Laboratory Medicine, Institute of Biomedical Sciences, Tokushima University Graduate School, Tokushima, Japan. [8]State Key Laboratory of Cancer Biology, National Clinical Research Center for Digestive Diseases and Xijing Hospital of Digestive Diseases, Air Force Military Medical University, Xi'an, China. [9]Division of Rheumatology, Allergy and Clinical Immunology, University of California Davis, Davis, CA, USA. [10]These authors contributed equally: Hao-Xian Zhu, Shu-Han Yang. ✉e-mail: liliang@gdph.org.cn; zzbin@mail.ustc.edu.cn; megershwin@ucdavis.edu; zxlian@gdph.org.cn

immune cell subsets and have identified a critical role of CD8[+] T cells in the development of biliary pathology[6–12].

The liver contains a specific immune microenvironment with a high proportion of resident immune cells, including tissue-resident memory CD8[+] T cells (Trm)[13,14]. CD8[+] Trm cells are a special subset that reside primarily in non-lymphoid tissues, function as a first line of defense, and respond rapidly upon pathogen rechallenge[15]. Most studies of CD8[+] Trm cells have focused on their role in immunity against pathogens. However, CD8[+] Trm cells also respond to self-antigens and mediate autoimmunity in many autoimmune diseases[16,17]. Their role in PBC remains to be further investigated.

In the treatment of PBC, as in other autoimmune diseases, the goal is to selectively deplete pathogenic cells. Chimeric antigen receptor (CAR)-T cells are designed to recognize and eliminate target cells expressing a specific antigen. CAR-T therapy has revolutionized oncotherapy[18]. Recently, anti-CD19 CAR-T cells have been studied in a murine model of systemic lupus erythematosus (SLE)[19,20], and the results of pilot studies of anti-CD19 CAR-T cells in human SLE are promising[21,22]. We report herein that CD8[+] Trm cells that highly express PD-1 mediate cytotoxicity against BECs in PBC and that PD-1-targeting CAR-T cells specifically deplete CD8[+] Trm cells in vivo and alleviate PBC in mice.

## Results

### CD8[+] T cells but not CD4[+] T cells are the primary contributors to murine autoimmune cholangitis

We previously reported that Il12b[−/−]Il2ra[−/−] (DKO) mice exhibit immunopathology similar to that in humans with PBC, including portal inflammation, bile duct damage, and liver fibrosis[12]. To investigate the pathogenic T cell subsets in our model, we crossed DKO mice with either Cd4[−/−] or Cd8a[−/−] mice. Deletion of Cd8a resulted in alleviated portal inflammation and bile duct damage (Fig. 1a–c). In contrast, liver histopathology was only slightly attenuated after deletion of Cd4 (Fig. 1a–c). The liver histology was consistent with these findings, with a decreased population of liver mononuclear cells (MNCs) in Cd8a[−/−]DKO mice, accompanied by significantly alleviated splenomegaly, which positively correlated with biliary inflammation[23] (Fig. 1d, e). Cd4 knockout also decreased liver MNC infiltration and spleen weight to some extent but to a lesser extent than Cd8a deletion (Fig. 1d, e). In addition, the expression levels of fibrosis-related genes (Acta2, Col1a1, Timp1, Tgfb1) and inflammation-related genes (Ifng, Ccl2, Tnf) were decreased in the livers of Cd8a[−/−] DKO mice but not in those of Cd4[−/−] DKO mice (Fig. 1f). The reduction of liver fibrotic area and the serum levels of aspartate aminotransferase (AST), alkaline phosphatase (ALP) was observed in Cd8a[−/−] DKO mice but not in those of Cd4[−/−] DKO mice (Supplementary Fig. 1a–c). These results indicate that CD8[+] T cells play a key role in murine cholangitis.

We also depleted CD8[+] T cells in DKO mice using an anti-CD8α antibody after significant biliary disease was noted at 8 weeks of age (Supplementary Fig. 1d, e). After CD8[+] T cell depletion, features of cholangitis, including portal inflammation and bile duct damage, in the livers of DKO mice were significantly improved (Fig. 1g–i). In addition, the number of liver infiltrated MNCs was reduced significantly, accompanied by attenuated splenomegaly (Fig. 1j, k). Fibrosis- and inflammation-associated gene expression was also downregulated after CD8[+] T cell depletion (Fig. 1l). Serological markers of liver function and fibrosis were improved after CD8[+] T cell depletion (Supplementary Fig. 1f–h). However, depletion of CD4[+] T cells using an anti-CD4 antibody did not lead to amelioration of autoimmune cholangitis (Supplementary Fig. 2a). The numbers of liver-infiltrating mononuclear cells and CD8[+] T cells did not change (Supplementary Fig. 2b), nor did liver histology (Supplementary Fig. 2c, d).

### Accumulation of tissue-resident memory CD8[+] T cells in the liver

To further understand the heterogeneity of liver-infiltrating CD8[+] T cells, we performed single-cell RNA-seq analysis of liver CD8[+] T cells

from DKO and WT mice. Unsupervised clustering identified 5 clusters of liver CD8[+] T cells (Fig. 2a). CD44 and CD62L (SELL) are markers used to define three major subsets of CD8[+] T cells in mice, namely, naïve T (Tnaive; Cluster 5; CD44[−]CD62L[+]), central memory T (Tcm; Cluster 2; CD44[+]CD62L[+]), and effector memory T (Cluster 1, 3; CD44[+]CD62L[−]) cells (Fig. 2a). Furthermore, we found that Clusters 1 and 3 represent distinct cell types with unique gene expression patterns (Fig. 2b). Cluster 1 had higher expression levels of cytokines, such as Gzma, Prf1 and Ccl5. In addition, Cluster 1 exhibited high expression of tissue residency-associated adhesion/retention molecules (Rgs1, Cxcr6, Itgal) and transcription factors (Bhlhe40, Runx3). Additionally, Cluster 1 exhibited reduced expression of circulating cell markers (Ly6c2, Il7r, Sell, Cx3cr1, Tcf7, Ccr7) (Fig. 2b and Supplementary Fig. 3a, b). By scoring the expression levels of up- and downregulated Trm signature genes, Cluster 1 was identified as Trm cells, and Cluster 3 was identified as Tem cells (Fig. 2c). In addition, Cluster 4, with high expression of the proliferation marker Ki67, was identified as proliferating T (Tproli) cells (Supplementary Fig. 3a, b). We found that Trm cells were enriched in DKO mouse livers compared to WT mouse livers (Fig. 2d). In contrast to the high proportion of naïve CD8[+] T cells in WT livers, the Tcm, Tem and Trm subsets were the major subsets of CD8[+] T cells in DKO mouse livers (Fig. 2d). Thus, we identified CD8[+] T-cell subsets with the residence marker CD69 and circulation marker CD62L by flow cytometry (Supplementary Fig. 3c). Consistent with the scRNA-seq data, a high proportion of CD69[+]CD62L[−]CD8[+] Trm cells accumulated in the livers of DKO mice but not in the spleens, liver draining lymph nodes (dLNs), or livers of WT mice (Fig. 2e, f). Moreover, CD8[+] Trm cells were found in the liver sinusoidal blood but not in the peripheral blood, portal venous blood or postcaval venous blood of DKO mice, indicating the liver residency of these cells[24] (Fig. 2g). DKO liver CD8[+] Trm cells exhibited a CXCR6[hi], CD11a[hi] and Ly6C[−] phenotype (Fig. 2h and Supplementary Fig. 3d); upregulation of Hobit and downregulation of S1pr1 (Fig. 2i). Parabiosis experiments confirmed the residency of DKO liver CD8[+] Trm cells comparing to peripheral blood CD8[+] T cells and liver CD8[+] Tcm cell subset (Fig. 2j, k). Developmental trajectory analysis of DKO liver CD8[+] T cells demonstrated that Tnaive cells developed along a trajectory passing through Tcm and Tem cells and terminally differentiated into Trm cells, suggesting that pathological factors may drive the differentiation of CD8[+] T cells into Trm cells in DKO livers (Supplementary Fig. 3e). Taken together, our results show that a distinct subset of CD69[+]CD62L[−]CD8[+] Trm cells resides and accumulates in the livers of DKO mice.

### Liver CD8[+] Trm cells from DKO mice exhibit increased activation and cytotoxicity

We further carried out detailed analyses of the phenotype and function of Trm cells in DKO livers. In comparison with those in Tcm and Tem cells, the expression levels of genes associated with TCR signaling, cytokines, and cytotoxicity in Trm cells were higher (Fig. 3a). By flow cytometry, we also detected high expression of cytotoxic molecules, including granzyme A, granzyme B, perforin, FASL and the degranulation marker CD107a, in Trm cells (Fig. 3b and Supplementary Fig. 3f). Trm cells demonstrated an exhausted phenotype, reflected by high expression of multiple immune checkpoint molecules (Fig. 3c), and this finding was confirmed by flow cytometry (Fig. 3d and Supplementary Fig. 3f). It is interesting that CD8[+] Trm cells possess both functional and exhaustion features. Accordingly, we analyzed CD8[+] Trm cell heterogeneity using scRNA-seq data and divided CD8[+] Trm cells into 3 subsets by unsupervised clustering (Fig. 3e). Among these 3 subsets, cells in Trm2 and Trm3 expressed Cd244a and could be distinguished by the expression of Cd200r1, while Trm1 expressed low levels of Cd244a and Cd200r1 (Fig. 3f). Developmental trajectory analysis identified Trm1 cells as the developmental origin cells, which transitioned into Trm2 cells and terminally differentiated into Trm3 cells (Fig. 3g). Gene expression profiling analysis also demonstrated

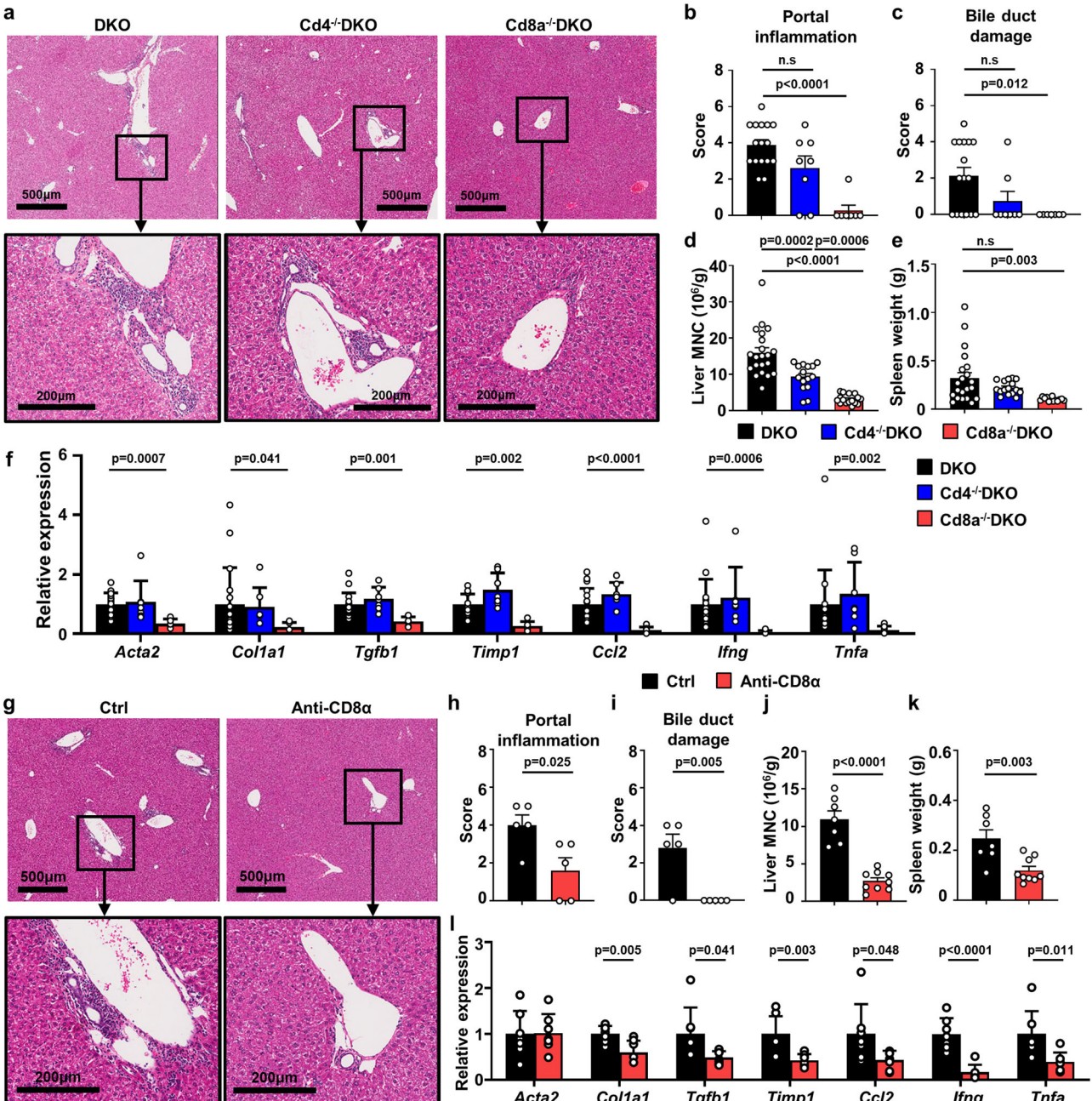

**Fig. 1 | CD8⁺ T but not CD4⁺ T cells are dominant in murine autoimmune cholangitis. a** Representative H&E staining pictures of liver from DKO, Cd4⁻/⁻DKO and Cd8⁻/⁻DKO mice, magnification showing the portal area. **b, c** Pathological score of portal inflammation and bile duct injury of liver of DKO (*n* = 16), Cd4⁻/⁻DKO (*n* = 8) and Cd8⁻/⁻DKO (*n* = 7). **d** Number of liver MNCs in DKO (*n* = 21), Cd4⁻/⁻DKO (*n* = 15) and Cd8⁻/⁻DKO (*n* = 16) mice. **e** Spleen weight of DKO (*n* = 21), Cd4⁻/⁻DKO (*n* = 16) and Cd8⁻/⁻DKO (*n* = 13) mice. **f** Relative mRNA expression level of inflammatory and fibrosis associated genes in the liver of DKO (*n* = 17), Cd4⁻/⁻DKO (*n* = 7) and Cd8⁻/⁻DKO (*n* = 7) mice. **g** Representative H&E staining pictures of liver from DKO mice treated with anti-CD8α depletion antibody and the control group, magnification showing the portal area. **h, i** Pathological score of portal inflammation and bile duct injury of DKO liver treated with anti-CD8α depletion antibody (*n* = 5) and the control group (*n* = 5). **j** Number of liver MNCs, and (**k**) spleen weight of DKO mice treated with anti-CD8α depletion antibody (*n* = 9) and the control group (*n* = 7). **l** Relative mRNA expression level of inflammatory and fibrosis associated genes in the liver of DKO mice treated with anti-CD8α depletion antibody (*n* = 7) and the control group (*n* = 7). The *p* values were determined by a one-way ANOVA with Tukey's multiple comparisons test (**b**–**f**), a two-tailed unpaired *t*-test (**h**–**l**). Data in (**a**, **g**) are representative results of at least two independent experiments. All experiments were repeated for 2–3 times. Data are shown as Means ± SEM.

that Trm1 cells expressed higher levels of stem-like-associated genes, exhibiting a precursor phenotype. Trm2 cells expressed higher levels of cytotoxicity- and cytokine-associated genes. Trm3 cells exhibited a terminally exhausted phenotype with high coinhibitory molecule expression, low costimulatory molecule expression and down-regulation of cytokine and cytotoxic molecule expression (Fig. 3h). By flow cytometry, Trm cells could be subclustered based on the

expression of 2B4 (name of the *CD244a* protein) and CD200R (Fig. 3i). The flow cytometry data also demonstrated that Trm1 was the precursor subset, with TCF-1 and CD127 expression; Trm2 was the hyperactivated and cytotoxic subset, with high production of IFN-γ and perforin; while the CD200R⁺Trm subset downregulated cytokine expression and increased apoptosis (Fig. 3j and Supplementary Fig. 3g). Thus, although CD8⁺ Trm cells express the exhaustion markers

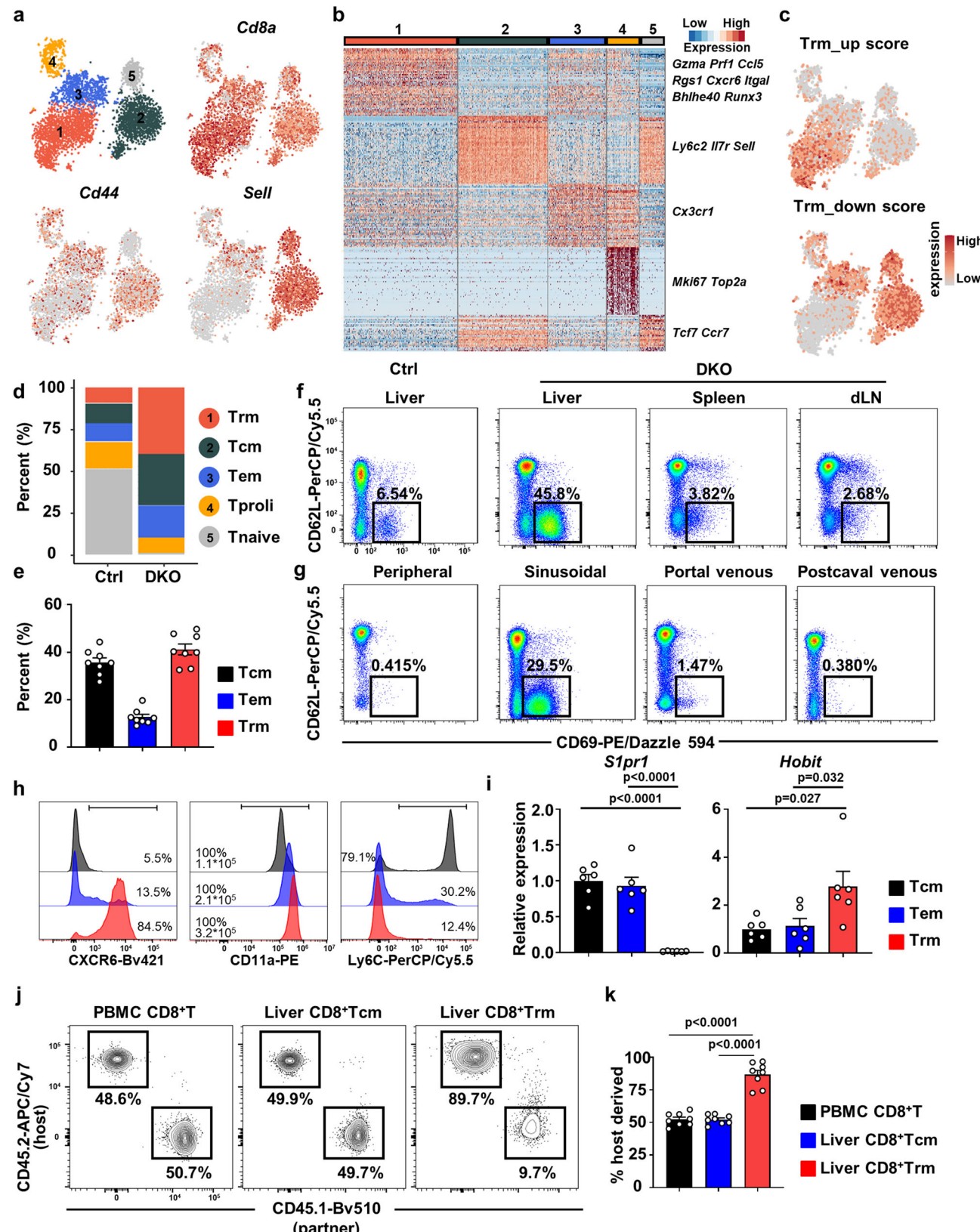

described in tumor T cells, they are activated and functional, and only a small proportion exhibit bona fide exhaustion. Importantly, Trm cells demonstrated a higher capability to kill primary bile duct epithelial cells than non-Trm cells in vitro (Fig. 3k, l).

In addition, similar cellular phenotype was observed in the liver of 2OA-BSA induced mice model and PBC patients. A large proportion of

CD8[+] T cells with residency features infiltrated in the liver of 2OA-BSA induced mice model (Supplementary Fig. 4a, b) and PBC patients (Supplementary Fig. 4g–j). In 2OA-BSA induced PBC model, liver CD8[+]Trm cells expressed high levels of cytotoxic molecules (Granzyme B, FasL) and inhibitory checkpoints (Tim-3, 2B4, CD160) (Supplementary Fig. 4c, d). Analogously, both cytotoxicity and exhaustion

**Fig. 2 | Accumulation of tissue-resident memory CD8⁺ T cells in liver of DKO mice. a** tSNE plots of WT and DKO liver CD8⁺ T cells, colored by cell clusters. Feature plots showing Cd8a, Cd44, Sell expression of all the CD8⁺ T cell clusters. **b** Expression heatmap shows top 50 differentially expressed genes of each cluster. Selected genes are indicated to the right, and complete lists of top genes are available in Supplementary Table 1. **c** Feature plots showing Trm up-or down-regulated genes expression of all the CD8⁺ T cell clusters. **d** Percentage of CD8⁺ cell clusters by Sc-RNA-seq data of WT and DKO mice. **e** Percentage of CD8⁺ cell clusters by flow cytometry of DKO mice (*n* = 8). **f** Representative FACS plot showing percentage of CD69⁺CD62L⁻CD8⁺ T cell subset from liver of Ctrl mice or liver, spleen and draining lymph node of DKO mice. **g** Representative FACS plot showing

percentage of CD69⁺CD62L⁻CD8⁺ T cell subset from peripheral, sinusoidal, portal venous and postcaval venous blood of DKO mice. **h** Representative FACS histogram of CXCR6, CD11a and Ly6C expression on CD8⁺ Tcm, Tem and Trm cell subsets. The ranged gate, the percentages of the positive population and gMFI of CD11a are shown in the figure. **i** Relative mRNA expression level of *S1pr1* and *Hobit* on CD8⁺ Tcm, Tem and Trm cell subsets in DKO mice liver (*n* = 6). Representative frequency plots (**j**) and statistical analysis (**k**) of peripheral blood CD8⁺ T cells or liver CD8⁺ Tcm and Trm cells derived from host (CD45.2⁺) or partner (CD45.1⁺) DKO mice (*n* = 8) after parabiosis. Data in (**f–h**, **j**) are representative results of at least three independent experiments. The *p* values were determined by a one-way ANOVA with Tukey's multiple comparisons test (**i**, **k**). Data are shown as Means ± SEM.

associated genes were enriched in liver CD8⁺ Trm cells compared to non-Trm cells in PBC patients (Supplementary Fig. 4k, l).

## PD-1 is a marker of highly activated liver CD8⁺ Trm cells in DKO mice

We then screened for markers for CD8⁺ Trm targeting by scRNA-seq analysis. Based on the differential gene expression profile between CD8⁺ Trm and non-Trm cells, Pdcd1 (encoding PD-1) is specifically highly expressed by CD8⁺ Trm cells (Fig. 4a, b). The flow cytometry data indicated that almost all CD8⁺ Trm cells expressed PD-1 and also expressed CXCR6 (Fig. 4c). Immunohistochemical staining demonstrated that CD8⁺ cells coexpressing CXCR6 and PD-1 infiltrated the portal area of DKO livers (Fig. 4d). Moreover, these PD-1⁺CD8⁺ cells were spatially closer to bile duct epithelial cells than were PD-1⁻CD8⁺ cells (Fig. 4e, f). Besides, PD-1 is also highly expressed in liver CD8⁺ Trm cells in 2OA-BSA induced PBC model (Supplementary Fig. 4e, f) and PBC patients (Supplementary Fig. 4m, n).

## PD-1-targeting CAR-T cells respond specifically to CD8⁺ Trm cells

We constructed a CAR to target Trm cells that highly and distinctively express PD-1. This CAR consists of the extracellular domain of PD-L1, the CD28 hinge and transmembrane domain, the CD28 intracellular domain, and a CD3ζ domain (Fig. 5a). A retroviral vector encoding the PD-1-targeting CAR was transduced into CD8⁺ T cells, and CAR-T cells were then sorted and expanded in vitro by stimulation with IL-7/IL-15 (Fig. 5b). CD8⁺ T cells from WT mice exhibited increased self-activation (Fig. 5c) and apoptosis (Fig. 5d) after CAR vector transduction, likely due to the intrinsic expression of PD-1 by CD8⁺ T cells. Therefore, we generated PD-1-targeting CAR-T cells using CD8⁺ T cells from PD-1-deficient mice. The specific reactivity of PD-1-targeting CAR-T cells was confirmed by upregulation of the activation marker CD69 in the presence of immobilized recombinant PD-1 protein (PD-1-FC), while the response was abolished by PD-L1 blockade (Fig. 5e). To further verify antigen-driven reactivity and cytotoxicity, we cocultured PD-1-targeting CAR-T cells with αCD3/CD28-induced PD-1⁺ T cells. PD-1-targeting CAR-T cells exhibited CD69 upregulation in response to PD-1⁺ T cells, and this effect was also eliminated by PD-L1 blockade (Fig. 5f). We also observed the proliferation of CAR-T cells after coculture with PD-1⁺ T cells (Fig. 5g). Importantly, PD-1-targeting CAR-T cells exhibited strong cytotoxicity against PD-1⁺ T cells (Fig. 5h, i) and liver CD8⁺ Trm cells from DKO mice (Fig. 5j), an effect that was also dependent on PD-1–PD-L1 binding.

## Depletion of liver CD8⁺ Trm cells by PD-1-targeting CAR-T cells ameliorates autoimmune cholangitis

We treated DKO mice at 8–10 weeks of age with PD-1-targeting CAR-T cells through intravenous infusion. PD-1 expression by peripheral T cells was examined to estimate the efficiency of CAR- therapy. PD-1⁺ T cells in the blood of DKO mice were depleted 1 week after CAR-T administration (Supplementary Fig. 5a, b), while the CAR-T cell percentage in blood increased (Supplementary Fig. 5c). Also, numbers of CAR-T cells infiltrated in DKO liver increased during the first week of treatment, declined rapidly in the second week and maintained at a low

cell number until the end point of treatment (Supplementary Fig. 5d). Nevertheless, cytokine release syndrome was not observed at the time (Supplementary Fig. 5e–i). The number of liver MNCs was significantly decreased after treatment (Fig. 6a), as were the percentages of PD-1⁺CD8⁺ T cells and CD8⁺ Trm cells (Fig. 6b, c). The absolute number of CD8⁺ Trm cells was significantly decreased with no alteration in the numbers of CD8⁺ Tcm and Tem cells (Fig. 6d–f). Liver inflammatory scores were also significantly reduced after CAR-T treatment (Fig. 6g, h). CD8⁺ T cells are dominant cell type within PD-1⁺ cells in DKO liver (Supplementary Fig. 5j). PD-1 targeting CAR-T cell infusion did not alter the cell number of NKT cells and macrophages and increased the cell number of NK cells which may due to decreased inflammation. The number of liver CD4⁺ T cells was also decreased after treatment (Supplementary Fig. 5k–o). To comprehensively investigate the therapeutic effect of PD-1-targeting CAR-T cells, we performed transcriptome sequencing on DKO livers after PD-1-targeting CAR-T cell or Ctrl-T cell infusion and the livers of healthy control mice. We found that compared to Ctrl-T cell infusion, CAR-T cell infusion significantly attenuated the pronounced alteration in gene expression in DKO livers (Fig. 6i). Indeed, Gene Ontology (GO) analysis showed that several immune response-related pathways were downregulated (Fig. 6j). Moreover, we evaluated the extent of liver inflammation and fibrosis remission by gene set enrichment analysis. Infusion of CAR-T cells resulted in decreases in the expression of genes upregulated in patients with PBC and genes associated with cholangitis in DKO livers. Additionally, CAR-T treatment effectively mitigated the gene set enrichment of fibrosis, inflammatory pathway and the IFN-γ response in DKO mouse livers (Fig. 6k). CAR-T treatment also led to ameliorated serum levels of ALT/ AST and the liver fibrosis (Supplementary Fig. 5p–r).

We also performed comprehensive analysis of the safety of PD-1 targeting CAR-T cells by infusing healthy control mice with PD-1-targeting CAR-T cells (Supplementary Fig. 6a). Compared to mice infused with Ctrl-T cells, the liver weight and the number of liver MNCs in mice infused with CAR-T cells showed no significant change (Supplementary Fig. 6b, c). CAR-T cell infusion had little effect on ALT and AST levels (Supplementary Fig. 6d, e). Histopathology of multiple organs showed no pathological changes after CAR-T cells infusion (Supplementary Fig. 6f–j). Furthermore, we assessed systemic immunity by the weight of the spleen and cytokine levels in the serum (Supplementary Fig. 6k–p). The data showed that PD-1 targeting CAR-T cell infusion have very low impact (Supplementary Fig. 6k, l). Together, detailed analysis showed that healthy mice infused with PD-1-targeting CAR-T cells have minimum effect of leading to inflammation or pathogenic changes in multiple organs.

## Discussion

We previously reported that CD8⁺ T cells are critical in mediating autoimmune cholangitis in murine models[8,9,11,25]. In this work, we utilized Il12b⁻/⁻Il2ra⁻/⁻ mice (DKO) and report that depletion of CD8⁺ T cells leads to almost complete remission of autoimmune cholangitis. The liver-infiltrating CD8⁺ T cell population contains a high proportion of highly activated CD8⁺ Trm cells, which mediate bile duct damage

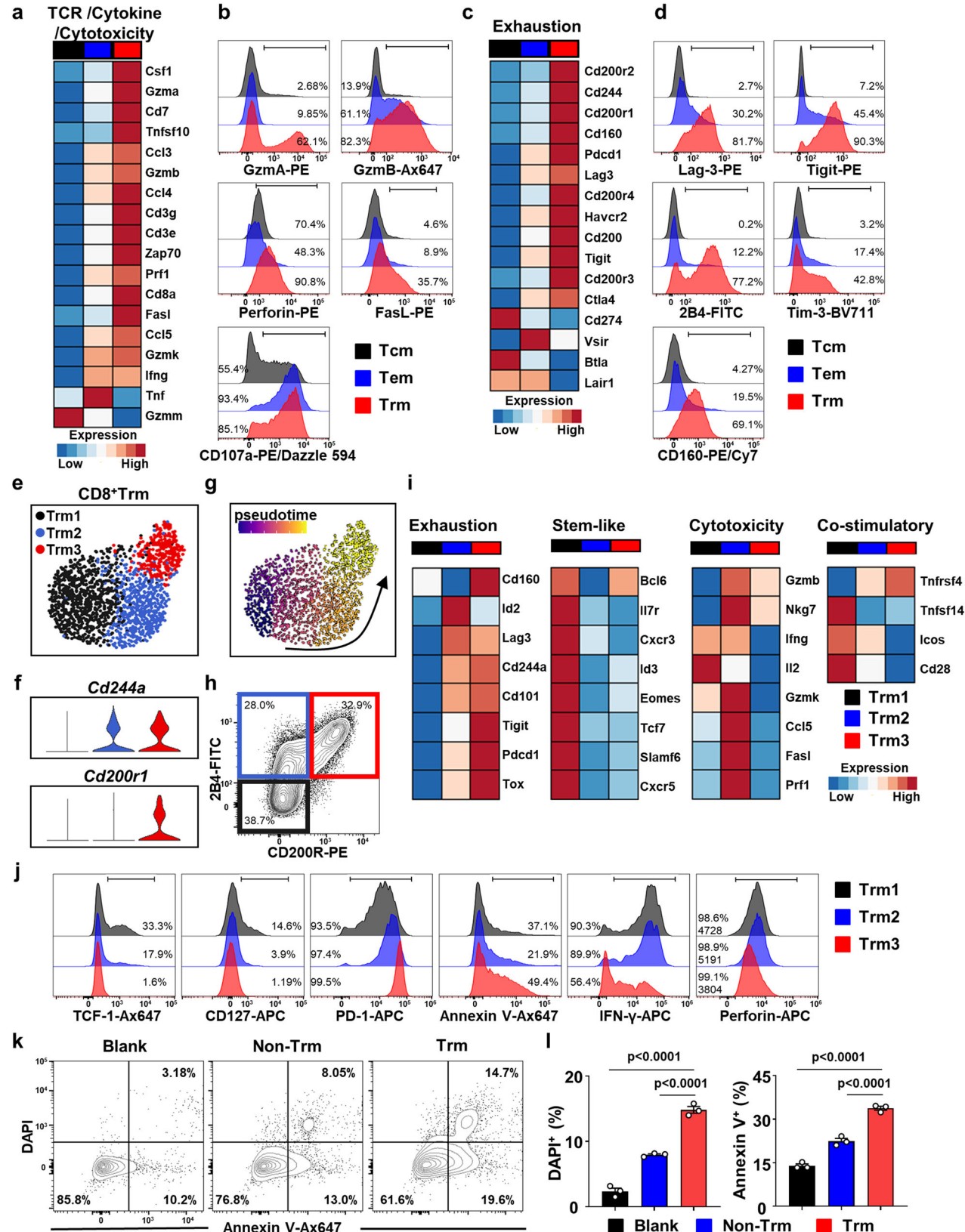

and disease progression. In addition, targeted depletion of CD8[+] Trm cells using PD-1-targeting CAR-T cells ameliorates autoimmune cholangitis. Our work highlights the use of PD-1-targeting CAR-T cells as a therapeutic strategy for PBC.

CD8[+] Trm cells have been reported to mediate several organ-specific autoimmune diseases[17,26–28], including autoimmune liver diseases[29,30]. However, unlike CD8[+] Trm cells in mucosal tissues, only a small portion of CD8[+] Trm cells in the liver express CD103. Our previous work demonstrated the expansion of CD103[+]CD8[+] Trm cells, which exhibit cytotoxicity against autologous cholangiocytes and are associated with clinical manifestations and UDCA response, in PBC patients[31]. We also observed a high proportion of CD69[+]CD8[+] Trm cells

**Fig. 3 | Liver CD8⁺ Trm cells from DKO mice exhibit enhanced activation and increased cytotoxicity. a** Heatmap of mRNA expression level of TCR signaling, cytokine production and cytotoxicity-associated genes of CD8⁺ Tcm, Tem and Trm cells from liver of DKO mice. **b** Representative FACS histogram of Granzyme A (GzmA), Granzyme B (GzmB), Perforin, FasL and CD107a expression on CD8⁺ Tcm, Tem and Trm cell subsets. The ranged gate and the percentages of the positive population are shown in the figure. **c** Heatmap of mRNA expression level of co-inhibitory moleculars associated genes of CD8⁺ Tcm, Tem and Trm cells from liver of DKO mice. **d** Representative FACS histogram of selected immune-checkpoint molecule expression on CD8⁺ Tcm, Tem and Trm cell subsets. The ranged gate and the percentages of the positive population are shown in the figure. **e** UMAP plots of DKO liver CD8⁺ Trm cells, colored by cell clusters. **f** Violin plots showing mRNA level of Cd244a and Cd200r1 of different Trm subset. **g** Differentiation trajectory of Trm subset by monocle analysis. Blue arrow indicating the probable direction of

differentiation. **h** Representative flow plots of CD200R and 2B4 expression on CD8⁺ Trm cells. **i** Heatmap of mRNA expression level of exhaustion, stemness, cyto-toxicity and costimulatory molecule associated genes of CD8⁺ Trm subsets. **j** Representative FACS plot of Trm clusters and representative FACS histogram graphs of TCF-1, CD127, PD-1, Annexin V, IFN-γ and Perforin expression on Trm subsets. The ranged gate, the percentages of the positive population and gMFI of Perforin are shown in the figure. **k** Representative flow plots of Annexin V and DAPI in PIBECs co-cultured with corresponding CD8⁺ T cell subsets. **l** Percentage of hepatic DAPI⁺ or Annexin V⁺ cells in PIBECs cocultured with corresponding CD8⁺ T cell subset from DKO mice liver. *n* = 3 cells examined over three independent experiments. Data in (**b, d, h, j**) are representative results of at least three independent experiments. The *p* values were determined by a one-way ANOVA with Tukey's multiple comparisons test (**l**). Data are shown as Means ± SEM.

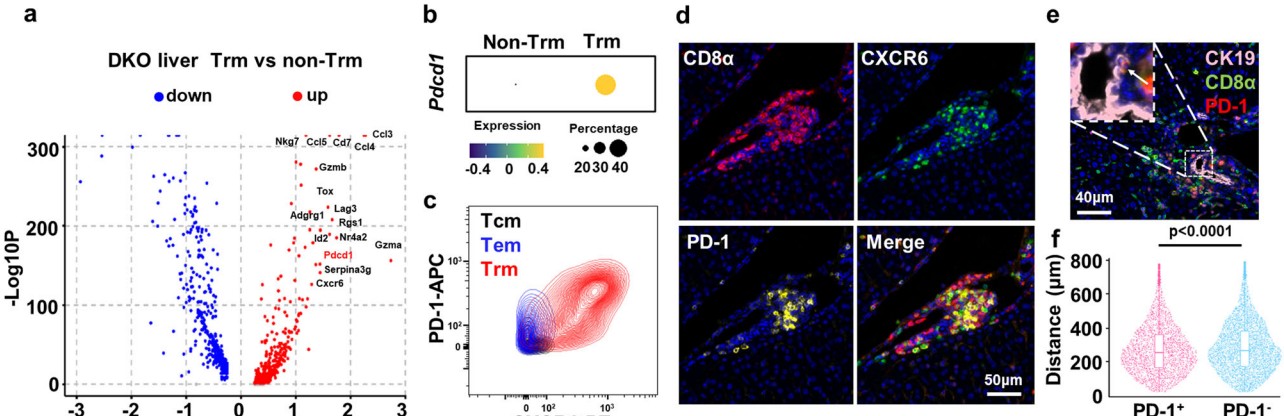

**Fig. 4 | PD-1 marks highly activated liver CD8⁺ Trm cells in PBC. a** Volcano plot shows the differentially expressed genes between CD8⁺ Trm and non-Trm cells in DKO liver. **b** Dot plot shows PD-1 expression of DKO liver CD8⁺ T cell subsets. The color of dots represents the average expression of genes and size for the percent of cells expressed the genes. **c** Representative flow plots of CXCR6 and PD-1 expression on DKO liver CD8⁺ Tcm, Tem and Trm cell subsets. **d** Representative multi-colored immunohistochemistry graph of CXCR6^GFP DKO liver stained with DAPI (blue), GFP (green), CD8 (red) and PD-1 (yellow). **e** Representative multi-colored immunohistochemistry graph of DKO liver stained with DAPI (blue), CK19 (pink), CD8 (green) and PD-1 (red). The white arrows pointed to PD-1⁺CD8⁺ T cells. **f** Statistics of the distance between PD-1^+/−CD8⁺ T cells and the nearest CK19⁺ bile duct epithelial cells in DKO liver (*n* = 3). Data in (**d, e**) are representative results of at least two independent experiments. Box-and-whisker plots show the median (center line), 25th, and 75th percentile (lower and upper boundary), whiskers extend to the farthest data points. The *p* values were determined by Wilcoxon test, Bonferroni *p* value correction (**a**), a two-tailed unpaired *t*-test (**f**).

in PBC patients. It has been reported that CD103⁻CD8⁺ Trm cells are true resident memory cells whose production is induced in inflammatory microenvironments and exhibit enhanced functional capacity compared to CD103⁺ Trm cells[32,33]. IL-15-induced liver-infiltrating CXCR6⁺CD103⁻CD8⁺ Trm cells are autoaggressive cells that mediate liver injury in human and mouse nonalcoholic steatohepatitis[34]. Herein, we found that CD8⁺ Trm cells in the DKO mouse model did not express CD103 or exhibit enhanced activation and cytotoxicity compared with effector memory CD8⁺ T cells, suggesting that they are also involved in the pathogenesis of PBC. However, the mechanism by which Trm cells selectively induce apoptosis of biliary epithelial cells in our model remains unknown. One of the possible reasons is that Trm cells are antigen-experienced and recognize autoantigen in biliary epithelial cells. We previously found that CD8⁺ Trm cells contain a high proportion of PDC-E2 specific T cells[31]. However, clonal expansion and antigen specificity of Trm cells need further investigation in the future. The other reason is that Trm cells, which express high level of cyto-toxic molecules as well as FASL and IFN-γ, infiltrate in the portal area and have a close spatial distance to biliary epithelial cells in PBC.

CD8⁺ Trm cells are antigen-experienced and highly express activation and cytotoxic molecules. However, we found that liver-infiltrating CD8⁺ Trm cells in PBC also expressed many immune checkpoint molecules, including markers of terminal exhaustion found in tumor-infiltrating T cells. Although activation-dependent exhaustion gene

expression is found in the setting of chronic infection and tumors[35,36], the pattern and mechanism of simultaneous activation and exhaustion in autoimmune diseases remain uncharacterized. Using scRNA-seq data, we found that liver CD8⁺ Trm cells are heterogeneous and identified a subset of cells with bona fide exhaustion that highly expresses the exhaustion marker CD200R. CD200R⁺CD8⁺ Trm cells lose expression of IFN-γ and undergo apoptosis. We also identified a differentiation trajectory of CD8⁺ Trm cells from stem-like cells to effector cells to terminally exhausted cells, suggesting an antigen-dependent manner of CD8⁺ Trm differentiation. In the future, we will analyze the TCR repertoire of these Trm cell subsets, which may provide insights for discovering autoantigen-specific T cells in PBC.

Targeting CD8⁺ Trm cells is a potential therapeutic strategy for autoimmune diseases. Studies have focused on blocking the development of CD8⁺ Trm cells by targeting IL-15, a cytokine necessary for CD8⁺ Trm cell generation and activation[28,37,38]. However, IL-15 blockade results in rapid depletion of NK cells and both CD4⁺ and CD8⁺ Tem cells in blood and tissues[39], suggesting that this strategy may not be suitable for clinical application. Moreover, IL-15 is dispensable for Trm maintenance or effector function in some situations[40–42]. In this work, we used a "living drug", CAR-T cells, to deplete pathogenic T cells in PBC. Anti-CD19 CAR-T cells have been studied in murine lupus, in which they resulted in a reduction in serum auto-antibodies and amelioration of kidney pathology[19,20]; also, the results

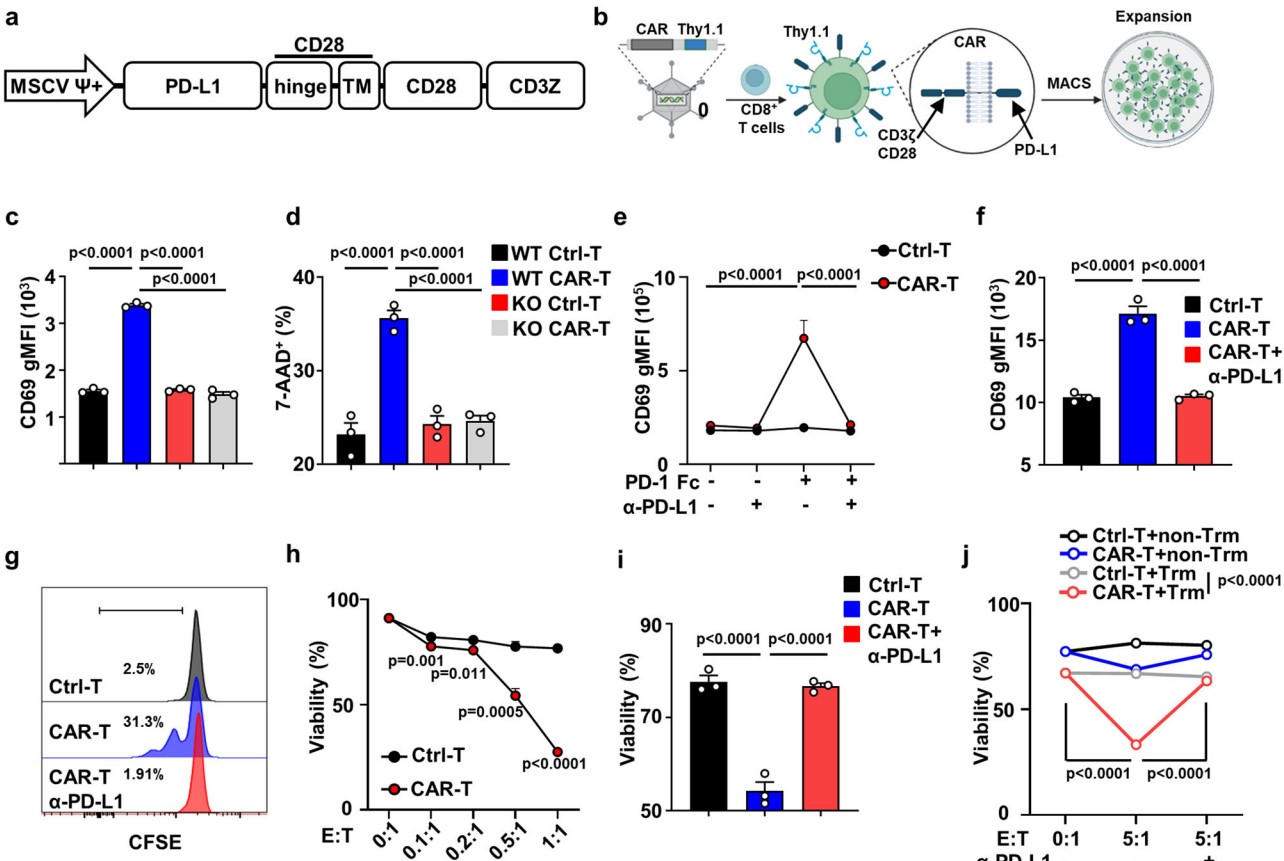

**Fig. 5 | PD-1 targeting CAR-T cells respond specifically to CD8+ Trm cells.**
**a** Schematic representation of the retroviral vector expressing the anti-PD-1 CAR. (TM transmembrane). The diagram was created with BioRender.com. **b** The experimental procedure of anti-PD-1 CAR-T preparation. **c** Level of CD69 expression 5 days after transduction. $n = 3$ cells examined over two independent experiments. **d** Percentage of 7-AAD+ cells 5 days after transduction. $n = 3$ cells examined over two independent experiments. **e** Level of CD69 expression of anti-PD-1 CAR-T and Ctrl-T cells under the stimulation of PD-1 FC and the blockade of anti-PD-L1 antibody. $n = 3$ cells examined over three independent experiments. Level of CD69 expression (**f**) and proliferation (**g**) of anti-PD-1 CART and Ctrl-T cells 1 day after coculture with αCD3/28 stimulated PD-1+ T cells with/without the PD-L1 blockade. The ranged gate and the percentages of the positive population are shown in the figure. $n = 3$ cells examined over three independent experiments. Cytotoxicity assay of anti-PD-1 CAR-T and Ctrl-T cells against (**h**, **i**) αCD3/28 stimulated PD-1+ T cells or (**j**) sorted Trm/non-Trm cells from liver of DKO mice at indicated effector/target ratio, with/without the PD-L1 blockade. $n = 3$ cells examined over three independent experiments. Data in (**g**) are representative results of three independent experiments. The $p$ values were determined by a one-way ANOVA with Tukey's multiple comparisons test (**c–f**, **i**, **j**), two-tailed unpaired $t$-test (**h**). Data are shown as Means ± SEM.

of anti-CD19 CAR-T therapy in SLE patients are encouraging[21,22]. We utilized the finding that CD8+ Trm cells highly express PD-1 and constructed a CAR containing the extracellular domain of the PD-1 ligand PD-L1. This CAR can enable CAR-T cells to respond to PD-1-expressing CD8+ Trm cells and exert cytotoxic effects as well as transduce suppressive signals through PD-1. PD-1-targeting CAR-T cells specifically decreased the number of CD8+ Trm cells without affecting other CD8+ T-cell subsets and relief the portal inflammation in DKO mice. We assume that PD-1-targeting CAR-T cells are also effective in other PBC models such as 2OA-BSA induced cholangitis and even PBC patients because of the similar observation that high expression of PD-1 on liver CD8+ Trm cells. Preclinical tests in different cholangitis models need to be performed for the future application of the therapy. Although healthy control mice infused with PD-1-targeting CAR-T cells develop undetectable inflammation or pathogenic changes in multiple organs, probably due to a low frequency of PD-1+ cells within organs at steady state, Trm cells do present in serval organs. Therefore, CAR-T cells may deplete Trm cells in other tissues, suggesting caution in their clinical application. Further efforts should focus on developing more specific CARs such as AND-gate CARs[43] with increased liver specificity for PBC.

CAR-T therapy combined with PD-1 blockade or engineered depletion PD-1 in CAR-T has been used to improve the therapeutic

efficiency in some clinical trials[44,45]. Recently, an enhanced type of CAR-T cells, which integrated an anti-CD19 CAR sequence into the PD1 gene, achieved high safety and efficacy in B cell non-Hodgkin lymphoma[46]. In this study, we generate anti-PD-1 CAR-T cells from CD8+ T cells derived from PD-1-deficient mice to prevent self-activation and fratricide of CAR-T cells. On the other hand, PD-1 deficiency also maintains CAR-T cell function and improve the potency of CAR-T cell therapies by interrupting PD-1/PD-L1 interaction. An optimized protocol to generate non-viral, gene-specific targeted CAR-T cells through gene editing can increase the feasibility of application using PD-1 targeting CAR-T cells in human.

Finally, PD-1-targeting CAR-T cells may be applied to other autoimmune diseases, as a PD-1-depleting antibody was found to ameliorate type 1 diabetes and multiple sclerosis in murine models[47]. Although depleting antibodies are much cheaper in clinical use, they face problems such as relapse due to incomplete depletion of target cells residing with in secondary lymphoid organs[48,49]. However, CAR-T cells can eliminate target cells both in the peripheral blood and within secondary lymphoid organs[50]. Moreover, CAR-T cells can persist in the host for a very long time for surveillance and induce sustained drug-free remission in patients[22]. Therefore, PD-1 targeting CAR-T cell therapy provides a strategy for patients that might resist a PD-1 depleting antibody therapy.

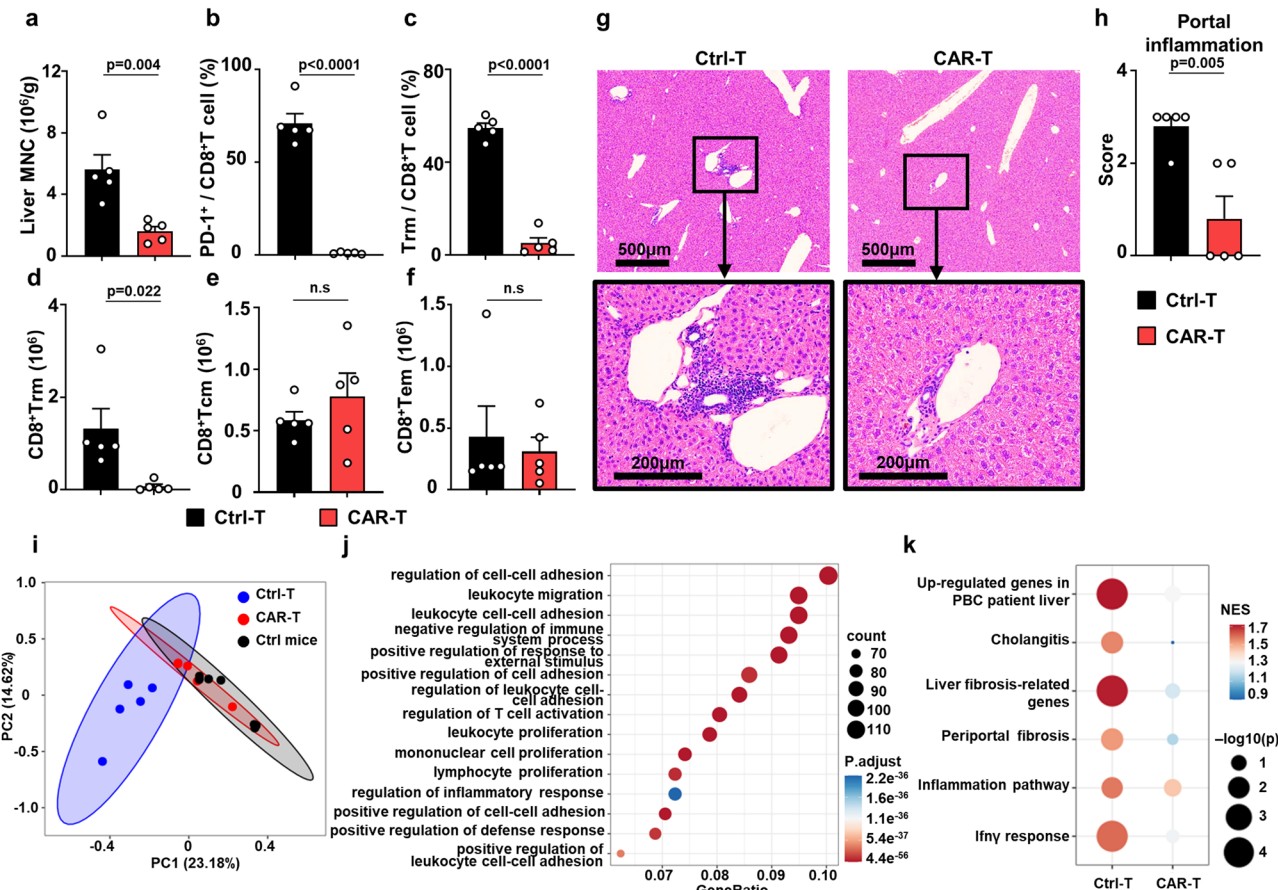

**Fig. 6 | Depletion of liver CD8⁺ Trm cells by PD-1 targeting CAR-T cells ameliorates autoimmune cholangitis. a** Number of hepatic mononuclear cells, (**b**) percentage of hepatic PD-1⁺CD8⁺T cells and (**c**) Trm cells in DKO mice after anti-PD-1 CAR-T ($n = 5$) or Ctrl-T infusion ($n = 5$). Absolute number of liver CD8⁺Trm (**d**), Tcm (**e**) or Tem (**f**) after anti-PD-1 CAR-T ($n = 5$) or Ctrl-T infusion ($n = 5$). **g** Representative H&E staining pictures of liver from DKO mice treated with anti-PD-1 CAR-T or Ctrl-T infusion, magnification showing the portal area. **h** Pathological score of portal inflammation of DKO liver after anti-PD-1 CAR-T ($n = 5$) or Ctrl-T infusion ($n = 5$). **i** PCA plots of liver transcriptome data of DKO mice after anti-PD-1 CAR-T ($n = 5$) or Ctrl-T infusion ($n = 5$) or ctrl mice ($n = 7$). **j** GO analysis of the upregulated genes in

DKO mice liver after anti-PD-1 CAR-T infusion comparing to Ctrl-T infusion. **k** Dot plot shows gene sets NES (normalized enrichment score) and $p$ value of DKO mice liver after anti-PD-1 CAR-T ($n = 5$) or Ctrl-T infusion ($n = 5$), comparing to ctrl mice liver respectively. The color of dots represents the NES score and size for −log10($p$ value). The $p$ values were determined by a two-tailed unpaired $t$-test (**a–f**, **h**). The $p$ values of GO analysis was generated in R using a Fisher's exact test (**j**). The $p$ values of GSEA was generated in GSEA software using an empirical phenotype-based permutation test (**k**). All experiments were repeated for 2–3 times. Data are shown as Means ± SEM.

## Methods

### Mice
Il2ra⁻/⁻ (B6.129S4-Il2ratm1Dw/J, #002952), Il12b⁻/⁻ (B6.129S1-Il12btm1Jm/J, #002693), Cd4⁻/⁻ (B6.129S2-Cd4tm1Mak/J, #002663), Cd8a⁻/⁻ (B6.129S2-Cd8atm1Mak/J, #002665), CXCR6ᴳᶠᴾ (B6.129P2-Cxcr6tm1Litt/J, #005693) and CD45.1 (B6.SJL-Ptprca Pepcb/BoyJ, #002014), and Pdcd1⁻/⁻ (B6.Cg-Pdcd1tm1.1Shr/J, #028276) mice on a C57BL/6J background were initially obtained from The Jackson Laboratory (Bar Harbor, ME, USA). Il12b⁻/⁻Il2ra⁻/⁻ mice were generated as previously reported, and Il12b⁻/⁻Il2ra⁺/⁻ mice were used as healthy control mice[12,51]. Cd4⁻/⁻Il12b⁻/⁻Il2ra⁻/⁻ mice, Cd8⁻/⁻Il12b⁻/⁻Il2ra⁻/⁻ mice, CD45.1 Il12b⁻/⁻Il2ra⁻/⁻ mice, and CXCR6ᴳᶠᴾIl12b⁻/⁻Il2ra⁻/⁻ mice were generated by backcrossing. Wild-type C57BL/6 mice were purchased from Hunan SJA Laboratory Animal Co., Ltd. (China). Induction of murine cholangitis by 2OA immunization are performed as previously described[52]. Briefly, Mouse was immunized with a mixture of 2-octynoic acid BSA conjugate (2OA-BSA) intraperitoneally in Complete Freund's Adjuvant (CFA, Sigma-Aldrich, St. Louis, MO) containing 10 mg/ml of Mycobacterium tuberculosis strain H37Ra and subsequently boosted every 2 weeks with 2OA-BSA and Incomplete Freund's Adjuvant (IFA) (Sigma-Aldrich). The mice studied herein were maintained in individually ventilated cages under specific pathogen-free

conditions in the Laboratory Animal Center, South China University of Technology. Phenotype analysis was performed on mice at 11–13 weeks of age regardless of sex. For Cd8a/Cd4 depletion or CAR-T therapy, mice were treated starting at 8 weeks of age and continuing to 12 weeks of age. 2OA-BSA induced mice were immunize at 6–8 weeks. For transgenic mouse models, experimental/control animals were co-housed. For induced mouse models, experimental/control animals were bred separately. Mice were sacrificed by cervical dislocation. Animal experiments were approved by animal ethics committee of South China University of Technology.

### Histology
Tissue sections were prepared and immediately fixed with 4% paraformaldehyde for 1–2 days. Tissues were embedded in paraffin and cut into 4-µm slices. All slices were deparaffinized and stained with hematoxylin and eosin or Sirus Red. The scores of liver inflammation and bile duct damage were assessed by pathologists in a blinded manner as previously reported[23].

### Cell isolation
Liver tissue was first homogenized with phosphate-buffered saline (PBS) containing 0.2% bovine serum albumin (BSA), passed through a steel

mesh, and resuspended. Mononuclear cells (MNCs) were isolated from suspended liver cells by centrifugation with 40% Percoll (GE Healthcare, Little Chalfont, United Kingdom). The spleen was disrupted by pressing between two glass slides, suspended in PBS/0.2% BSA, and passed through a 70-micron nylon mesh. Red blood cells were depleted using RBC lysis buffer (Beyotime, China), and cells were suspended in PBS and counted on a hemocytometer based on the presence of trypan blue. Surgical methods to collect portal venous, postcaval venous and liver sinusoidal blood were previously described[24].

## Flow cytometry
For flow cytometry, cell suspensions were incubated with a purified anti-CD16/CD32 antibody (BioLegend, San Diego, CA, USA) for 15 min at 4 °C and then stained for 20 min at 4 °C with cocktails containing combinations of fluorochrome-conjugated monoclonal antibodies specific for cell surface markers. For intracellular cytokine staining, cells were resuspended in RPMI-1640 medium supplemented with 10% fetal bovine serum and stimulated with Cell Stimulation Cocktail (plus protein transport inhibitors) (eBioscience, San Diego, CA, USA) at 37 °C for 4 h. After surface marker staining, the cells were fixed and permeabilized with a Cytofix/Cytoperm ™ Fixation/Permeabilization Kit (BD Biosciences, San Jose, CA, USA) and then stained for intracellular IFN-γ with an anti-IFN-γ antibody (BioLegend). For transcription factor staining, normal IgG isotype controls (BioLegend) were used in parallel. A BD LSR Fortessa (BD Immunocytometry Systems, San Jose, CA, USA) was used to acquire data, which were analyzed with FlowJo software (Tree Star, Inc., Ashland, OR, USA). Antibodies specific for the following proteins were used in this study: Perforin-APC (1:200, Clone S16009A, Cat#154303, BioLegend); PD-1-APC (1:400, Clone 29F.1A12, Cat#135209, BioLegend); Ly-6C-APC (1:1000, Clone HK1.4, Cat#128015, BioLegend); IFN-g-APC (1:200, Clone XMG1.2, Cat#505809, BioLegend); CD127 (IL-7Ra)-APC (1:400, Clone A7R34, Cat#135011, BioLegend); Ly-6C-APC/Cy7 (1:1000, Clone HK1.4, Cat#128025, BioLegend); CD45.2-APC/Cy7 (1:200, Clone 104, Cat#109823, BioLegend); TCF1-Ax647 (1:200, Clone C63D9, Cat#6709S, CST); GZMB-Ax647 (1:200, Clone QA16A02, Cat#372219, BioLegend); Annexin V-Ax647 (1:50, Clone, Cat#640911, BioLegend); CD8α-Ax700 (1:1000, Clone 53-6.7, Cat#100729, BioLegend); CD69-BV421 (1:200, Clone H1.2F3, Cat#104527, BioLegend); CD45.2-BV510 (1:200, Clone 104, Cat#109837, BioLegend); CD45.1-BV510 (1:200, Clone A20, Cat#110741, BioLegend); TCRb-BV605 (1:200, Clone H57-597, Cat#109241, BioLegend); CD44-BV711 (1:1000, Clone IM7, Cat#103057, BioLegend); CD8α-BV785 (1:1000, Clone 53-6.7, Cat#100749, BioLegend); CD4-BV785 (1:400, Clone RM4-5, Cat#100551, BioLegend); CD44-FITC (1:1000, Clone IM7, Cat#103021, BioLegend); CD244.2-FITC (1:400, Clone m2B4 (B6)458.1, Cat#133503, BioLegend); PD-1-PE (1:400, Clone 29F.1A12, Cat#135205, BioLegend); CD200R (OX2R)-PE (1:200, Clone OX-110, Cat#123907, BioLegend); CD178 (FasL)-PE(1:50, Clone MFL3, Cat#106605, BioLegend); CD11a-PE (1:400, Clone M17/4, Cat#101107, BioLegend); GZMA-PE (1:200, Clone 3G8.5, Cat#149704, BioLegend); Perforin-PE (1:200, Clone S16009A, Cat#154305, BioLegend); Lag-3 (CD223)-PE (1:200, Clone C9B7W, Cat#12-2231-82, eBioscience); TIGIT (Vstm3)-PE (1:200, Clone 4D4/mTIGIT, Cat#156103, BioLegend); CD160-PE (1:200, Clone 7H1, Cat#143003, BioLegend); NK1.1-PE/Cy5 (1:400, Clone S17016D, Cat#156523, BioLegend); PD-1-PE/Dazzle 594 (1:400, Clone 29F.1A12, Cat#135227, BioLegend); CD69-PE/Dazzle 594 (1:400, Clone H1.2F3, Cat#104535, BioLegend); CD107a (LAMP-1)-PE/Dazzle 594 (1:400, Clone 1D4B, Cat#121623, BioLegend); Ly-6C-PerCP/Cy5.5 (1:1000, Clone HK1.4, Cat#128011, BioLegend); CD62L-PerCP/Cy5.5 (1:1000, Clone MEL-14, Cat#104431, BioLegend); TIM3-PE/Cy7 (1:400, Clone RMT3-23, Cat#119715, BioLegend).

## Quantitative real-time PCR
Total RNA was extracted from liver tissues using TRIzol Reagent (Invitrogen, Carlsbad, CA). M-MLV Transcriptase (Invitrogen) was used for reverse transcription. Quantitative real-time PCR was performed with an AB StepOne real-time PCR system (Applied Biosystems, Carlsbad, CA) using Premix Ex Taq (Takara, Japan). The PCR primers used herein are listed in Supplementary Table 2.

## Depletion of CD8+/CD4+ T cells in vivo
First, 200 μg of *InVivoMAb* anti-mouse CD8α (BioXCell, Clone: 2.43) or anti-mouse CD4 (BioXCell, Clone: GK1.5) was injected *i.p.* three times weekly starting at 8 weeks of age and continuing to 12 weeks of age. *InVivoMAb* Rat IgG2b Isotype Control (BioXCell, Clone: LTF-2) was used as a control.

## CAR construction and retroviral production
The extracellular domain sequence of PD-L1 was designed to be fused to the CD28 hinge and transmembrane domains, the CD28 costimulatory domain and the CD3Z signaling domain.

The CAR sequence above was inserted into a pMSCV retroviral vector containing Thy1.1 as the selection marker (pMSCV-IRES-Thy1.1) by ligation-independent cloning to construct the pMSGV-CAR-IRES-Thy1.1 plasmid. DNA sequencing was used to confirm the expected sequence. An Endo-free Plasmid Midi Kit (Omega Bio-tek) was used to produce high-quality DNA. The retroviral vector encoding the PD-1-targeting CAR or the empty vector was transduced into HEK293T cells along with the pCL-Eco retrovirus packaging vector. Lipofectamine 8000™ (Beyotime) was used as the transfection reagent. Forty-eight hours after transduction, the virus-containing medium was collected and filtered through a 0.45 μm filter.

## Manufacturing of CAR-T cells
Murine CD8+ T cells from WT mice or Pdcd1−/− mice were enriched from single-cell suspensions of dissociated spleens with CD8 (TIL) MicroBeads (Miltenyi Biotec) by magnetic-activated cell sorting (MACS). Purified CD8+ T cells were activated with plate-bound anti-CD3 (4 μg/ml) and anti-CD28 antibodies (2 μg/ml) in RPMI 1640 medium containing FCS (10%), sodium pyruvate (1 mM), penicillin–streptomycin (100 U/ml), β-mercaptoethanol (0.1 mM) and IL-2 (100 U/ml) for 2 days in 12-well plates. On Day 3, the viral supernatant was transferred to RetroNectin-coated 12-well plates (TaKaRa). Activated CD8+ T cells were transduced by spin infection at 800 × g for 90 min[53]. One day after transduction, the virus-containing medium was replaced with fresh medium supplemented with 100 U/ml IL2, and the cells were cultured for two more days. Then, Thy1.1+ cells were positively selected with CD90.1 MicroBeads (Miltenyi Biotec) by MACS. Purified transduced T cells were further supplemented with IL-7 and IL-15 without IL-2 as described in the method for T-cell expansion[54]. On Day 7, cells were harvested for follow-up experiments. Control T cells were processed with the same methods but transduced with empty vector.

## In vitro assays
For the CAR-T cell-specific lysis assays, anti-CD3/28-induced PD-1+ T cells were used as target cells. Briefly, CD8+ T cells were isolated from the spleens of WT mice and incubated with plate-bound anti-CD3 (BioLegend, 4 μg/ml) and anti-CD28 (BioLegend, 2 μg/ml) antibodies in the T-cell medium described above for 48 h. Then, anti-CD3/28 antibody-induced PD-1+ T cells (5 × 10^4 cells/well) were cocultured with CAR-T cells or Ctrl-T at the indicated effector:target ratio for 24 h. Freshly isolated Trm and non-Trm cells from DKO mouse livers were cocultured with CAR-T or Ctrl-T cells at an effector:target ratio of 1:5 for 12 h. An anti-PD-L1 blocking antibody (BioLegend, 5 μg/ml) was added as indicated in the figures.

For the Trm cytotoxicity assays, freshly isolated CD8+ Trm and CD8+ non-Trm cells from DKO mouse livers were cocultured with primary bile duct epithelial cells (Procell Life Science & Technology Co., Ltd.) at an effector:target ratio of 3:1 for 12 h. Cells were collected, stained and analyzed by flow cytometry. Cell viability was assessed by Annexin V staining.

T-cell proliferation was assessed by labeling with carboxy-fluorescein succinimidyl ester (CFSE) (2 μM, Life Technologies, Waltham, MA, USA). Briefly, CFSE-labeled CAR-T cells or Ctrl-T cells were added to the coculture system with PD-1⁺CD8⁺ T cells.

For the recombinant PD-1 stimulation assay, CAR-T cells were activated with plate-bound Mouse PD-1-Fc Chimera (BioLegend, 10 μg/ml) in T-cell medium.

### CAR-T therapy in vivo
DKO mice were injected twice with $1 \times 10^6$ CAR-T or Ctrl T cells via the tail vein, once at 8 weeks of age and once at 10 weeks of age. Mice were sacrificed 4 weeks after the first treatment.

### RNA sequencing
Total RNA was extracted with an EZ-press RNA Purification Kit (EZBioscience, MN, USA). For RNA sequencing, mRNA was isolated using VAHTS mRNA Capture Beads (Vazyme, Nanjing, China), and sequencing libraries were constructed using the VAHTS Universal V8 RNA-seq Library Prep Kit for Illumina (Vazyme). Sequencing reads were aligned to the mouse reference genome GRCm38 using STAR-2.5.2b software. Data were analyzed using custom Perl and R scripts. In brief, the read counts of each gene were converted to reads per kilobase million (RPKM) values. RNA sequencing analysis was performed according to a standard procedure as previously described[55]. Compared with the livers of DKO mice infused with Ctrl-T cells, the upregulated genes (log2fc > 1) in the livers of DKO mice infused with CAR-T cells were subjected to Gene Ontology analysis using the clusterProfiler package (version 4.2.2) in R[56]. Gene Set Enrichment Analysis (GSEA) software (version 4.2.3) downloaded from MSigDB was used to assess whether specific signatures were significantly enriched in DKO mouse livers after anti-PD-1 CAR-T or Ctrl-T cell infusion compared to ctrl mouse livers[57]. The Cholangitis (HP_CHOLANGITIS), Periportal fibrosis (HP_PERIPORTAL_FIBROSIS), Inflammation pathway (BIOCARTA_INFLAM_PATHWAY), and Ifnγ response (HALLMARK_INTERFERON_GAMMA_RESPONSE) gene sets were downloaded from the Molecular Signatures Database (MSigDB). Upregulated genes in PBC patient livers and liver fibrosis-related genes were previously reported[58,59].

Raw sequencing datasets were deposited into the NCBI Gene Expression Omnibus under accession number GSE241014.

### Single-cell capture, cDNA synthesis and library preparation
Single-cell RNA sequencing was performed according to a standard procedure as previously described. In brief, single-cell capture was achieved by Poisson distribution of a single-cell suspension across >200,000 microwells (BD Rhapsody™). Beads with oligonucleotide barcodes were added to saturation to pair with the beads with the cells in the microwells[60]. Cell lysis buffer was added to lyse the cells to allow capture of polyadenylated molecules, including mRNAs, targets and sample tags, with the barcode-conjugated beads. Reverse transcription was performed on the beads for WTA and sample tag library preparation. Finally, the library was sequenced using the NovaSeq platform (Illumina, San Diego, CA) to generate 150 bp paired-end reads.

### Single-cell RNA sequencing data analysis
We downloaded the singe-cell RNA sequencing data of DKO mice from NCBI Gene Expression Omnibus (GSE186333). The raw sequencing data for gene expression and extracellular protein expression were aligned with the mm10 mouse reference genome using the STAR algorithm in the BD Rhapsody analysis pipeline (version 1.9.2β, BD). We removed cells with more than 4000 or less than 200 unique genes or more than 10% mitochondrial genes. Finally, we obtained 13,695 cells for analysis, specifically, 6897 cells isolated from DKO mice and 6798 cells isolated from WT mice. The data matrix was transformed as a Seurat object with the Seurat package (version 4.0.1). Batch effects between the datasets were removed with CCA in the Seurat package.

The Monocle package was used to infer the developmental trajectory of CD8⁺ T cells. Differentially expressed genes were identified using the FindMarkers function in Seurat using Wilcoxon's test, the Bonferroni correction, and 0.25 as the log (fold change) threshold. Significantly differentially expressed genes with a $p$ value of <0.01 are labeled red (upregulated) or blue (downregulated). GSEA was performed using the ClusterProfiler package (version 3.16.1). Preranked genes were used in GSEA to calculate $p$ values and enrichment scores. The Trm upregulation and downregulation scores used in this study were previously described[61,62].

We downloaded the singe-cell RNA sequencing data of PBC patients from Genome Sequence Archive (HRA002347). Data were pre-processed and quality control was performed as previously described[63]. Briefly, individual cells were filtered with a UMI cutoff value of <30,000 and gene value between 200 and 4000, and the cells with mitochondrial gene content >50% were deleted. The gene sets used in this study are reported previously[27,36].

### Multiplex immunohistochemical staining and data analysis
Multiplex immunohistochemical staining was performed following the protocol of the Opal Polaris 7-Color Manual IHC Kit (Akoya Bioscience, NEL861001KT). Antibodies against mouse CD8α (1:800, D4W2Z), F4/80 (1:200, D2S9R) and GFP (1:400, D5.1) were purchased from Cell Signaling Technology. Antibodies against mouse PD-1 (1:100, EPR20665), and CK19 (1:1500, EP1580Y) were purchased from Abcam. Slides were imaged, whole slides were scanned using a Vectra Polaris Automated Quantitative Pathology Imaging System (Akoya Biosciences), and multispectral images were acquired using Phenochart software, version 1.0.12 (Akoya Biosciences), to unmix and remove autofluorescence. The distances between CD8⁺ cells and CK19⁺ cells were calculated by HALO software (Indica Labs).

### Serum biochemistry
Enzymatic activities of aspartate aminotransferase (AST) and alanine aminotransferase (ALT) in serum were evaluated by Biochemistry Automatic Analyzer 3100 (Hitachi Ltd) using diagnostic kits from Shanghai Kehua Bio-engineering Co., Ltd.

### Cytometric bead array
Serum concentrations of IFN-γ, TNF-α, IL-6, IL-10 and MCP-1 from DKO mice or control mice treated with CAR-T were measured with a cytometric bead array kit (Mouse Inflammation Kit, BD Biosciences) according to the manufactories' instructions.

### Parabiosis
Parabiosis surgery was performed as previously described[64]. A pair of CD45.1 and CD45.2 DKO mice were anesthetized. Longitudinal skin incisions were performed from the elbow to the knee joint. Then, connect the skin of the two animals with a continuous 5-0 Vicryl stapling. Blood chimerism was confirmed by cytometry analysis of CD45.1+/ CD45.2+ cells in periphery blood. Mice were sacrificed after blood completely exchange.

### Statistics
Statistical significance was analyzed using GraphPad Prism 8 (Graph-Pad Software, San Diego, CA, USA). The results in all figures are expressed as the means ± standard error of the means. For two-group comparisons, statistical analyses were performed using a two-tailed unpaired $t$-test with Welch's correction. For multiple group comparisons, statistical analyses were performed using one-way ANOVA followed by Tukey's multiple comparison test. The $p$ value of differential genes were identified using the FindMarkers function from Seurat using Wilcoxon test, Bonferroni $p$ value correction and 0.25 as logfc.threshold. $p$ values <0.05 were considered to indicate significant differences.

**Reporting summary**

Further information on research design is available in the Nature Portfolio Reporting Summary linked to this article.

## Data availability

The RNA sequencing data in this study were deposited in the Gene Expression Omnibus database under accession number GSE241014. Source data are provided with this paper.

## Code availability

The data analysis pipeline in our study is described on the BD Rhapsody analysis pipeline website, Seurat official website and clusterProfiler official website. The code can be found at https://github.com/zhuhx97/DKO.git.

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

## Acknowledgements

This work is supported by the National Natural Science Foundation of China (82120108013 to Z.X.L., 82270551 to Y.H., 81820108005 to Y.H., 82370528 to Z.B.Z., 82001730 to S.H.Y.). We thank Prof. Penghui Zhou (Sun Yat-sen University Cancer Center) for helping with CAR construction and inspiring discussions. We thank Prof. Liming Nie (Guangdong Provincial People's Hospital, Guangdong Academy of Medical Sciences, Southern Medical University) for helping with 2-OA BSA induced PBC mouse model construction. We thank Prof. Jin Chai (Southwest Hospital, Third Military Medical University (Army Medical University)) for providing scRNA-seq data of PBC patients. Various graphical schematics were created using BioRender.

## Author contributions

Study conception and design: H.X.Z., S.H.Y., L.L., Z.B.Z., M.E.G. and Z.X.L. Analyzed data and wrote the manuscript: H.X.Z. and SHY. Acquisition of data: H.X.Z., S.H.Y., C.Y.G., X.M.C. and R.R.H. Edited and revised the manuscript: X.L., Q.L.M., H.S.J., Y.H., L.L. and Z.B.Z. Access the pathology score: K.T. Funding acquisition and supervision: S.H.Y., C.Y.G., Q.L.M., Y.H., L.L., Z.B.Z., M.E.G. and Z.X.L. Acquired and analyzed data in revision: H.X.Z., C.Y.G. and Z.H.B.

## Competing interests

The authors declare no competing interests.
