## [Peer Review File · Nature Communications]

REVIEWER COMMENTS

Reviewer #1 (expert in autoimmunity, CAR T cells):

The manuscript by Zhu et al. characterises the cytotoxic autoimmune response in a mouse model for PBC described by the authors earlier in 2014 and not frequently used by other researchers in the field. Primarily by scRNA analysis infiltrating CD8 cells were characterised to contain CD8 T cells with expression profiles overlapping with tissue resident memory T cells (TRM), displaying both effector (granzymes, perforin) but also exhaustion characteristics. Not surprisingly a large fraction of these cells express the PD1 molecule. The authors then target PD1 positive cells by a novel CAR-T strategy, which was able to ameliorate T cell infiltrations significantly.

These findings are novel and significant and offer new therapeutic developments for autoimmune liver diseases. The CD8 cells might very well resemble in their phenotype the recently described auto-aggressive CXCR6+ CD8 T cells causing liver immune pathology in NASH.

I have two major concerns that should be addressed in a revised version of the manuscript:

1. Liver infiltrating lymphocytes should be characterised by flow cytometry in a much deeper way and ideally compared to well-established TRM populations in infectious disease and exhausted populations in cancer models or chronic infections. This is important for comparing the observed population to other disease model systems and even to human data.

2. PD1-targeting CAR-T therapy must be characterised in more detail, particularly side-effects of the therapy in other organs are documented only rudimentarily. Specifically, analysis of CAR-T cell infiltrations in the liver but also other organs should be quantitatively analysed over time.

The statement "PD-1 targeting T cells did not exhibit inflammation ... in multiple organs" is not sufficiently supported by the data presented. In fact, histology suggests a significant colitis which is entirely uncommented. As the authors write in the last two sentences depletion of Trm/PD-1 expressing T cells in other tissues might have significant impact. Therefore, more detailed analyses are mandatory.

The necessity to use CD8 cells from PD1-deficient mice could be evaluated in the discussion with the potential of gene editing of CAR-T cells for a potential application in humans.

Reviewer #2 (expert in hepatology and liver immunology):

In this study, the authors used p40^{-/-}/IL2Ra^{-/-} (DKO) mice to demonstrate the pathological role of CD8Trm cells in murine PBC model. Genetic CD8 T cell depletion and killing antibodies ameliorated PBC-like

cholangiopathy in this model; treatment with PD-1-targeted CAR-T cells, which can deplete systemic PD-1 positive cells including CD8^{Trm} cells, also improved cholangiopathy. The data is solid and the findings are potentially interesting, however, there are many issues that need to be addressed.

Major:

1. It remains unclear whether CD8^{Trm} cells contribute as the main cells to disease progression in this model. While the significance of using PD-1-targeted CAR-T cells is acknowledged, a more comprehensive analysis of the cell population in the host immune system during this treatment should be performed. Although the authors only demonstrated Trm specificity among T cells, the diversity of PD-1 expressing cells requires a detailed examination of other immune cells. In addition, other methods to selectively eliminate Trm cells should be explored to clarify this important point. Anti-CXCR3 or anti-CXCR6 antibodies may be useful.
2. What is the mechanism by which Trm cells selectively induce apoptosis of biliary epithelial cells in this model?
3. The authors should investigate the role of CD8^{Trm} and the effect of PD-1 targeting CAR-T cells in other models as well. It is assumed that evaluation using this model, which lacks the major proliferation and differentiation factors of CD8 T cells, may not reflect the human condition. Evaluation using inducible models in wild-type mice (e.g. DDC or 2-OA) is recommended.
4. To prove that the increased CD8^{Trm} cells in this model are CD8^{Trm}, it is essential to examine the parabiosis model. The gene and protein expression patterns of these cells indicate that they are Trm-like, but this data is insufficient.
5. Data necessary for evaluation of cholangitis are lacking. Serological markers (e.g., ALP, ALT, BIL) and staining of fibrotic sites (e.g., MT or SR) are needed in Figures 1 and 6.
6. What is the reason for the increase in CD8^{Trm} in the absence of IL-12 and IL-2 in this model, could IL-15 explain it? If so, proof should be provided using anti-IL-15 neutralizing antibodies.
7. The reason for choosing CAR over PD-1 antibody should be discussed. Is it because CAR can remove cells from tissues more efficiently than antibodies? Unless superiority is demonstrated, CAR treatment will be avoided as it is extremely expensive in clinical use.

Author Response to the Editor's and Reviewers' Comments

Reviewer #1

The manuscript by Zhu et al. characterises the cytotoxic autoimmune response in a mouse model for PBC described by the authors earlier in 2014 and not frequently used by other researchers in the field. Primarily by scRNA analysis infiltrating CD8 cells were characterised to contain CD8 T cells with expression profiles overlapping with tissue resident memory T cells (TRM), displaying both effector (granzymes, perforin) but also exhaustion characteristics. Not surprisingly a large fraction of these cells express the PD1 molecule. The authors then target PD1 positive cells by a novel CAR-T strategy, which was able to ameliorate T cell infiltrations significantly.

These findings are novel and significant and offer new therapeutic developments for autoimmune liver diseases. The CD8 cells might very well resemble in their phenotype the recently described auto-aggressive CXCR6⁺ CD8 T cells causing liver immune pathology in NASH.

I have two major concerns that should be addressed in a revised version of the manuscript:

1. Liver infiltrating lymphocytes should be characterised by flow cytometry in a much deeper way and ideally compared to well-established TRM populations in infectious disease and exhausted populations in cancer models or chronic infections. This is important for comparing the observed population to other disease model systems and even to human data.

Thank you for your advice.

We compared DKO CD8⁺ Trm cells to CD8⁺ Trm cells in other models as suggested (Please see the figure below). We found that CD8⁺ Trm cells in liver cancer (Hepa1-6 cell-induced orthotopic liver cancer model) express high levels of exhaustion markers (Lag-3, Tigit, 2B4, Tim-3, CD160, PD-1) as previously reported (a). However, tumor infiltrating CD8⁺ Trm cells are bona fide exhaustion cells with decreased expression of effector molecules Granzyme B, CD107a, Perforin, FasL and IFN- γ (b). Moreover, we also compared the phenotype CD8⁺ Trm cells in DKO mice to CD8⁺ Trm cells in tumor and infectious conditions using scRNA-seq data. We analyzed CD8⁺ T cell data from mouse models of LCMV infection (c, d), in vivo expression of HBV antigen by rAAV8-1.3HBV (e, f) and Hepa1-6 cell-induced orthotopic liver cancer model (g, h) (Cell Rep. 2020 Aug 25;32(8):108078, J Infect Dis. 2019 Feb 15;219(5):750-759, Nat Commun. 2023 Dec 9;14(1):8154). We found CD8⁺ Trm cells exist in different models, as identified by high Cxcr6 expression and low Sell expression (c, e, g). Similar to CD8⁺ Trm cells from DKO mice, CD8⁺ Trm cells from other models also highly express cytotoxic and exhaustion molecules (d, f, h). However, CD8⁺ Trm cells from DKO liver have the highest expression of cytokine Ifng and cytotoxic molecules Gzmb and Prf1 (i). This can be explained that only a small proportion of DKO liver CD8⁺ Trm cells are bona fide exhausted as shown in figure 3. These data suggest that CD8⁺ T cells in DKO liver may have a pathogenic role. Relative studies of deeply comparing liver CD8⁺ Trm cells in different disease conditions are ongoing in our lab.

2. PD1-targeting CAR-T therapy must be characterised in more detail, particularly side-effects of the therapy in other organs are documented only rudimentarily. Specifically, analysis of CAR-T cell infiltrations in the liver but also other organs should be quantitatively analysed over time.

The statement "PD-1 targeting T cells did not exhibit inflammation ... in multiple organs" is not sufficiently supported by the data presented. In fact, histology suggests a significant colitis which is entirely unmentioned. As the authors write in the last two sentences depletion of Trm/PD-1 expressing T cells in other tissues might have significant impact. Therefore, more detailed analyses are mandatory.

The necessity to use CD8 cells from PD1-deficient mice could be evaluated in the discussion with the potential of gene editing of CAR-T cells for a potential application in humans.

Thanks for your valuable suggestion.

Similar to the rapid expansion of CAR-T cells in peripheral blood in the first week after infusion, the number of CAR-T cells infiltrated in DKO liver increased during the first week of treatment, declined rapidly in the second week and maintained at a low cell number until the end point of treatment (Supplementary figure 5d).

Also, we performed more comprehensive analysis of the safety of PD-1 targeting CAR-T cells in Supplementary figure 6. CAR-T cell infusion has little effect on liver weight, liver MNC, serum aspartate aminotransferase (AST), alkaline phosphatase (ALP) (Supplementary figure 6a-e). Histopathology of several organs were added and show no pathological changes after CAR-T cells infusion (Supplementary figure 6f). Moreover, the body weight, colon length and the colon weight-length ratio show no difference between mice infused with CAR-T and Ctrl-T cells (Supplementary figure 6g-j). We also deposited the detailed histology of the colon of three mice from the CAR-T and Ctrl-T group respectively, with the comment from our pathologist Koichi Tsuneyama in a blinded way (Please see the figure below). Furthermore, CAR-T cell infusion did not induce systemic inflammation as no changes were observed in spleen weight (Supplementary figure 6k) and serum inflammatory cytokine levels (Supplementary figure 6l-p). Together, detailed analyses showed that WT mice infused with PD-1-targeting CAR-T cells have minimum effect of leading to inflammation or pathogenic changes in multiple organs.

We also added discussion about the necessity to use CD8 cells from PD1-deficient mice in our revised manuscript line 341-350.

In this study, the authors used p40^{-/-}IL2Ra^{-/-} (DKO) mice to demonstrate the pathological role of CD8^{Trm} cells in murine PBC model. Genetic CD8 T cell depletion and killing antibodies ameliorated PBC-like cholangiopathy in this model; treatment with PD-1-targeted CAR-T cells, which can deplete systemic PD-1 positive cells including CD8^{Trm} cells, also improved cholangiopathy. The data is solid and the findings are potentially interesting, however, there are many issues that need to be addressed.

Major:

1. It remains unclear whether CD8^{Trm} cells contribute as the main cells to disease progression in this model. While the significance of using PD-1-targeted CAR-T cells is acknowledged, a more comprehensive analysis of the cell population in the host immune system during this treatment should be performed. Although the authors only demonstrated Trm specificity among T cells, the diversity of PD-1 expressing cells requires a detailed examination of other immune cells. In addition, other methods to selectively eliminate Trm cells should be explored to clarify this important point. Anti-CXCR3 or anti-CXCR6 antibodies may be useful.

Thanks for your valuable suggestion. We agree that selectively deplete Trm cells in DKO mice can provide solid evidence to prove the pathogenic role of CD8⁺ Trm cells. However, we lack the means to achieve this purpose. In vivo anti-CXCR6 blockade antibody is unavailable and CXCR3 blocking may influence many cell types, including CD8⁺ Tem/ Tem, Th1, B cells and NK cells, and we have demonstrated that knockout of CXCR3 in DKO mice exacerbated cholangitis (Front Immunol. 2018 May 17;9:1090). On the other hand, our data shows that CD8⁺ T cells are the dominant cell type within PD-1⁺ cells in DKO liver (Supplementary figure 5j). Nonetheless, we excluded the pathogenic role of CD4⁺ T cells in our model by CD4 depletion. Herein, anti-PD-1 CAR-T therapy, which is relatively specific to deplete CD8⁺ Trm cells, provide evidence for CD8⁺ Trm cells acting as the main cells that contribute to disease progression.

We also added the examination of other immune cells in DKO liver after anti-PD-1 CAR-T therapy in Supplementary figure 5. Anti-PD-1 CAR-T do not alter the cell number of NKT cell and macrophage and increase the cell number of NK cell which may due to decreased inflammation. Although CD4⁺T cell number decreased, they do not participate in cholangitis in our model. We recognize the shortcoming of this therapy and are committed to develop a more precise therapy by CAR-T optimizing. Ways to improve the CAR-T specificity were added in the discussion.

2. What is the mechanism by which Trm cells selectively induce apoptosis of biliary epithelial cells in this model?

Thank you for the critical comment. Although we demonstrated Trm cells are pathogenic cells and therapeutic targets in PBC, the mechanism by which Trm cells selectively induce apoptosis of biliary epithelial cells in our model remains unknown. One of the possible reasons is that Trm cells are antigen-experienced and recognize autoantigen in biliary epithelial cells. We previously found that CD8⁺ Trm cells contain a high proportion of PDC-E2 specific T cells (J Hepatol. 2022 Nov;77(5):1311-1324.). However, clonal expansion and antigen specificity of Trm cells need further investigation in the future. The other reason is that Trm cells, which express high level of cytotoxic molecules as well as FASL and IFN- γ , infiltrate in the portal

area and have a close spatial distance to biliary epithelial cells in PBC.

3. The authors should investigate the role of CD8^{Trm} and the effect of PD-1 targeting CAR-T cells in other models as well. It is assumed that evaluation using this model, which lacks the major proliferation and differentiation factors of CD8 T cells, may not reflect the human condition. Evaluation using inducible models in wild-type mice (e.g. DDC or 2-OA) is recommended.

Thanks for your valuable suggestion. We have previously described the role of CD8^{Trm} cells, especially CD103⁺CD8^{Trm} cells in PBC patients. Also, we show that these cells have a higher PD-1 expression in PBC patients than healthy control (J Hepatol. 2022 Nov;77(5):1311-1324.). We also observed an increase of CD8^{Trm} cells in 2-OA mouse model (Supplementary figure 4a). CD8^{Trm} cells from 2-OA mouse liver have a similar phenotype to DKO mice, highly expressing tissue resident marker (CXCR6, CD11a, Ly6C), cytotoxic molecules (GzmB, FasL) and exhausted markers (Tim3, 2B4, CD160), regardless of lack of IL-2 and IL-12 signal in DKO CD8^{Trm} cells (Supplementary figure 4b-d). Moreover, we found a large population of CD8^{Trm} cells infiltrated in liver of PBC patients, making use of a recently published single cell RNA-seq data (Nat Commun. 2023 Feb 9;14(1):29) (Supplementary figure 4g-j). Similar enrichment of both cytotoxic and exhaustion genes in CD8^{Trm} cells were observed (Supplementary figure 4k-l). More importantly, CD8^{Trm} cells from both 2OA-BSA model and PBC patients expressed higher levels of PD-1 (Supplementary figure 4e-f, m-n), suggesting that PD-1 targeting CAR-T cells may also be applied in 2OA-BSA induced PBC model and potentially in human. The lack of evaluation in different models as a limitation of this study were discussed in our revised manuscript (line 329-335).

4. To prove that the increased CD8T cells in this model are CD8^{Trm}, it is essential to examine the parabiosis model. The gene and protein expression patterns of these cells indicate that they are Trm-like, but this data is insufficient.

Thanks for your valuable suggestion. We performed parabiosis experiment and confirmed that DKO liver CD8^{Trm} cells were overwhelmingly noncirculating, as shown in Figure 2k in our revised manuscript. However, there are a small proportion of partner derived CD8^{Trm} cells. We assumed that these cells are differentiated from periphery counterpart rapidly due to the severe inflammation in DKO liver.

5. Data necessary for evaluation of cholangitis are lacking. Serological markers (e.g., ALP, ALT, BIL) and staining of fibrotic sites (e.g., MT or SR) are needed in Figures 1 and 6.

Thank you for your professional advice. We have added the serum level of ALT and AST and the Sirius red staining in Supplementary figure 1 (a-c, f-h) and 5 (p-r). Both the serological markers and fibrosis level are attenuated in CD8a^{-/-}DKO or DKO treated with anti-CD8a antibody or anti-PD-1 CAR-T cells.

6. What is the reason for the increase in CD8T in the absence of IL-12 and IL-2 in this model, could IL-15 explain it? If so, proof should be provided using anti-IL-15 neutralizing antibodies.

Thank you very much for your professional and detailed suggestions. As one of the important factors in Trm differentiation, previous study minimizes liver CD8^{Trm} cells in NASH model by interrupting IL-15 signal (Nature. 2021 Apr;592(7854):444-449). However,

seral study also shows that IL-15 are dispensable for Trm maintain or effector function (J Immunol (2016) 196 (9): 3920–3926. Eur J Immunol. 2015 Dec;45(12):3324-38. Science. 2023 Dec;382(6674):1073-1079). We found that IL-15 neutralizing antibody treatment reduced MNCs infiltrated in the DKO liver (a), with reduced percentage (b) and number (c) of CD8⁺ Trm cells

In our previous study, we found that deficiency of IFN- γ improves liver inflammation in DKO mice (J Immunol Res. 2022 Mar 7:2022:7111445). Although IFN- γ KO did not alter the absolute number of CD8⁺ T cells in DKO liver, it decreased the density of CD8⁺ T cells (d). Also, lack of IFN- γ resulted in significant decrease in the percentage (e) and number (f) of CD8⁺ Trm cells in DKO liver. Taken together, IFN- γ may promote the hyperactivation of the CD8⁺ T cells and differentiation of CD8⁺ Trm cells in a direct or indirect manner. Moreover, lots of cytokine pathways are upregulated in DKO liver. It is assumed that cytokines collaboration promotes the inflammation. Deficiency in one cytokine can be compensated by another cytokine in inflammation environments, which can explain why we delete seral pro-inflammatory cytokines (IL-18, IL-21) in DKO mice and did not attenuate the portal inflammation. In conclusion, multiple cytokines may participate in CD8⁺ T cell hyperactivation and CD8⁺ Trm cell differentiation in our PBC model.

7. The reason for choosing CAR over PD-1 antibody should be discussed. Is it because CAR can remove cells from tissues more efficiently than antibodies? Unless superiority is demonstrated, CAR treatment will be avoided as it is extremely expensive in clinical use.

We thank the reviewer for this critical comment. Although depleting antibodies are much cheaper in clinical use, they face problems such as relapse due to incomplete depletion of target cells residing with in secondary lymphoid organs (Am J Transplant. 2013 Jun;13(6):1503-11; Ann Rheum Dis. 2022 Jan;81(1):100-107). However, CAR-T cells can eliminate target cells both in the peripheral blood and within secondary lymphoid organs. (Blood. 2016 Jul 21;128(3):360-70). Moreover, CAR-T cells can persist in the host for a very long time for surveillance and induce sustained drug-free remission in patients (Nat Med. 2022

Oct;28(10):2124-2132). Therefore, PD-1 targeting CAR-T cell therapy provides a strategy for patients that might resist a PD-1 depleting antibody. We have added discussions about this in the revised manuscript (line 353-359).

REVIEWERS' COMMENTS

Reviewer #1 (Remarks to the Author):

The revised manuscript by Zhu et al. significantly improved. My two major remarks regarding the characterisation of the CD8+ TRM population as well the better characterisation of the CAR-T treatment regarding kinetics of effector cells and safety aspects are adequately addressed in the revised manuscript, and I congratulate the authors for this interesting and important manuscript.

Reviewer #2 (Remarks to the Author):

The authors have adequately addressed my concerns and the paper is much improved. On the other hand, I still believe that it is important to show the effect of PD-1 targeting CAR-T cells in 2-OA mouse model used as a second model in order to demonstrate their universal utility in this area.